# Endothelial cell-specific loss of eNOS differentially affects endothelial function

**Shuhan Bu[1], Hien C. Nguyen[1,2], Sepideh Nikfarjam[1,2], David C. R. Michels[1], Berk Rasheed[1,2], Sauraish Maheshkumar[1], Shweta Singh[1], Krishna K. Singh**[1,2]*

1 Department of Medical Biophysics, Schulich School of Medicine and Dentistry, University of Western Ontario, London, Ontario, Canada, 2 Anatomy and Cell Biology, Schulich School of Medicine and Dentistry, University of Western Ontario, London, Ontario, Canada

* krishna.singh@uwo.ca

**Data Availability Statement:** All relevant data are contained within the paper and its Supporting Information files.

## Abstract

The endothelium maintains and regulates vascular homeostasis mainly by balancing interplay between vasorelaxation and vasoconstriction *via* regulating Nitric Oxide (NO) availability. Endothelial nitric oxide synthase (eNOS) is one of three NOS isoforms that catalyses the synthesis of NO to regulate endothelial function. However, eNOS's role in the regulation of endothelial function, such as cell proliferation and migration remain unclear. To gain a better understanding, we genetically knocked down eNOS in cultured endothelial cells using sieNOS and evaluated cell proliferation, migration and also tube forming potential *in vitro*. To our surprise, loss of eNOS significantly induced endothelial cell proliferation, which was associated with significant downregulation of both cell cycle inhibitor p21 and cell proliferation antigen Ki-67. Knockdown of eNOS induced cell migration but inhibited formation of tube-like structures *in vitro*. Mechanistically, loss of eNOS was associated with activation of MAPK/ERK and inhibition of PI3-K/AKT signaling pathway. On the contrary, pharmacologic inhibition of eNOS by inhibitors L-NAME or L-NMMA, inhibited cell proliferation. Genetic and pharmacologic inhibition of eNOS, both promoted endothelial cell migration but inhibited tube-forming potential. Our findings confirm that eNOS regulate endothelial function by inversely controlling endothelial cell proliferation and migration, and by directly regulating its tube-forming potential. Differential results obtained following pharmacologic *versus* genetic inhibition of eNOS indicates a more complex mechanism behind eNOS regulation and activity in endothelial cells, warranting further investigation.

## Introduction

The endothelium forms an innermost layer of every blood vessel and acts as a macromolecular barrier that regulates movement of small and large molecules, paracrine factors and gases between blood vessels and adjacent tissues [1]. The endothelial cells also modulate platelet adherence, leukocyte migration, smooth muscle cell proliferation and apoptosis [1]. In addition, they modulate blood fluidity by regulating platelet production and control inflammatory responses by producing cytokines and adhesion molecules [2]. One of the major function of

**Funding:** Krishna Singh received funding from the Project Grant (FRN # 153216), Canadian Institutes of Health Research, Canada to KS. KS is also the recipient of the 2018/19 National New Investigator Award- Salary Support from the Heart and Stroke Foundation of Canada, Canada.

**Competing interests:** The authors have declared that no competing interests exist.

endothelium is to regulate vascular tone by balancing the production of flow-sensitive nitric oxide (NO), which is an endothelium-dependent vasodilator of the underlying smooth muscle cells [3]. Under normal physiological conditions, NO has cardioprotective effects, where it diffuses to the surrounding tissues and results in muscle relaxation and prevents inflammation by inhibiting leukocyte adhesion and platelet aggregation [3]. Kubes *et al.*, have shown that inhibition of NO in the vascular endothelium results in upregulation of CD11/CD18, protein essential for leukocyte adhesion [4]. Additionally, NO is also critical for the process of vascular remodelling, specifically VEGF-induced angiogenesis. It is reported that angiogenesis is proportional to the level of NO production in mouse vessels [5].

Inhibition of NO production occurs in the cardiovascular diseases such as atherosclerosis [6]. The shortage of NO is thought to occur either by decreased NO production or increased NO degradation [7]. NO is produced by an enzyme called endothelial nitric oxide synthases (eNOS). This enzyme catalyzes the conversion of L-Arginine into L-Citrulline, producing NO in the process [8]. Therefore, decreased NO production can be attributed to decreased availability of L-arginine or decreased availability and activity of eNOS [8]. ENOS, is expressed mainly in endothelium of the large arteries and impaired eNOS expression is implicated in development and progression of cardiovascular diseases [9]. ENOS and NO production constitutes a positive feed-back loop; upon changes in blood flow NO production increases and activates PI3-kinase dependent phosphorylation of protein kinase B leading to phosphorylation of eNOS and production of NO [10]. NO can also directly modulate expression of eNOS; such as inhibiting endogenous NO production using nitro-L-arginine-methyl ester (L-NAME) results in reduction of eNOS expression and increased exogenous NO causing increased eNOS expression [11].

In summary, eNOS is an essential regulator of endothelial function that constitute angiogenic potential, proliferation and migration. There are several reports investigating the role of eNOS and NO on endothelial angiogenic potential. Accordingly, Namba *et al.*, reported that eNOS overexpression promotes angiogenesis and stimulates vascular endothelial growth factor (VEGF) production, and that L-NAME, a pharmacologic inhibitor of eNOS, inhibits eNOS-overexpression-associated blood flow [12]. There are other reports demonstrating inhibition of angiogenesis following knockdown of eNOS *in vivo* [5, 13]. There are also reports on cell proliferation following pharmacological inhibition of eNOS by its inhibitor L-NAME but this is not investigated following genetic loss of eNOS in endothelial cells. Therefore, to better delineate the role of eNOS in endothelial function and to understand related mechanisms, we genetically inhibited eNOS in cultured endothelial cells and investigated endothelial function and related regulators of endothelial function *in vitro*. We for the first time show that knockdown of eNOS significantly induces endothelial cell proliferation, downregulates both cell cycle inhibitor p21 and cell proliferation antigen Ki-67; induces migration and inhibits tube forming potential of endothelial cells. Loss of eNOS significantly activate MAPK/ERK, but inhibits PI3-K/AKT signaling pathway in endothelial cells. As previously reported, pharmacologic inhibition of eNOS inhibited cell proliferation and migration. Differential results obtained following pharmacologic *versus* genetic inhibition of eNOS indicates a more complex mechanism behind eNOS regulation and activity in endothelial cells, warranting further investigations.

## Material and methods

### Cell culture

Human umbilical vein or human pulmonary artery endothelial cells (HUVECs or HPAECs, pooled, Lonza; passage 4–6) were cultured in endothelial cell growth medium-2 (EGM™-2

Bulletkit$^{TM}$; Lonza) supplemented with growth factors, serum and antibiotics at 37˚C in humidified 5% $CO_2$. siRNA-mediated eNOS gene knockdown was performed with sieNOS or scrambled control in accordance with the manufacture's guidelines. A standard reverse transfection reagent (Lipofectamine ® 3000, Invitrogen), and 5 nm sieNOS [Dharmacon$^{TM}$: Cat # 106158 (sieNOS) or #106159 (sieNOS#)] or scrambled control (Dharmacon$^{TM}$: Cat # 2575450) were used. RNA or protein were collected using Trizol reagent (Invitrogen) or RIPA buffer, respectively, following 24, 48 or 72 hrs post-transfection.

## Quantitative real time PCR

Total RNA was extracted, and complementary DNA (cDNA) was synthesized using 1μg RNA and the QuantiTech Reverse Transcription Kit (Qiagen). SYBR Select Master Mix (Applied Biosystems), forward and reverse primers for eNOS or GAPDH [14], and cDNA were mixed, and qPCR was performed using QuantStudio®3 Real-Time PCR instrument (Applied Biosystems). Comparative Delta Delta CT method was employed for data analysis.

## Immunoblotting

Cell lysates from cultured endothelial cells were prepared in RIPA buffer (Sigma) 24, 48, and 72 hrs post-transfection with either sieNOS or its scrambled control. Equal amounts of protein were loaded on SDS-polyacrylamide gels and transferred to PVDF membrane (Thermo Fisher). Membranes were blocked for 1 hour and incubated with primary antibody specifically targeting eNOS, Ki67, p21, AKT, p-AKT, ERK, and p-ERK overnight at 4˚C. After incubating with secondary antibody, bands were visualized with ECL substrate using chemiluminescence channel and 700 channel in Li-Cor Fc Odyssey imaging system, and quantified.

## Proliferation assay

HUVECs were first cultured in endothelial cell growth medium-2 (EGM$^{TM}$-2 Bulletkit$^{TM}$; Lonza) supplemented with growth factors, serum and antibiotics at 37˚C in humidified 5% $CO_2$, and seeded at a density of $1-2\times10^4$ cells/well in 96-well plates, transfected with sieNOS (5 nmol) or scrambled control (5 nmol), and then cell proliferation was evaluated 24, 48 and 72 hrs post-transfection using WST-8 Cell Proliferation Assay Kit (Cayman Chemicals) as described [15]. HUVECs were transfected in a 6-well plate ($1.5-2\times10^5$ cells/well) with either sieNOS or scrambled control and cells were counted using CytoSmart cell counter 24, 48 and 72 hrs post-transfection. Cells were counted in triplicates for each biological replicate and average cell number was determined for each sample.

## Scratch assay

HUVECs were cultured in endothelial cell growth medium-2 (EGM$^{TM}$-2 Bulletkit$^{TM}$; Lonza) supplemented with growth factors, serum and antibiotics at 37˚C in humidified 5% $CO_2$ and transfected with sieNOS (5 nmol) or scrambled control (5 nmol), and seeded at a density of 2 x $10^5$ cells/well in a 6-well plate and allowed to grow to 60–80% confluency. Each well was then administered a consistent straight scratch using p1000 pipette tip. Cells were then washed with 1X PBS for one time and incubated in DMEM supplemented with 1% FBS. Phase-contrast microscopy using an adapted camera (Optika) was employed to take pictures of cells in each well migrating into the scratch over 3 time points (0, 8 and 20 hrs) to evaluate for migrating capacity as described [16].

## In vitro tube-formation assay

The *In vitro* Angiogenesis Kit (Millipore) was employed to evaluate endothelial angiogenic properties. HUVECs were transfected with sieNOS and seeded at a density of 2 x $10^5$ cells/well in a 6-well plate and allowed to grow to ~75% confluency. The kit-provided matrix solution was added into designated wells of a 96-well plate. Transfected cells from the previous preparation were then harvested and seeded at a density of 1–1.5 x $10^4$ cells/well onto the designated wells in EGM-2 medium. Phase-contrast microscopy was employed (Optika) to take pictures of cells under phase-contrast in each designated well over time to monitor formation of tube-like structures *in vitro*. Mesh area and tube-length was measured using angiogenesis analyzer software from Image J.

## Statistical analysis

Data are expressed as the mean ± SD. Student's *t-test* was applied when the means of two groups were compared using GraphPad-Prism software. A p-value <0.05 was considered statistically significant.

## Results and discussion

In order to understand the role of eNOS in endothelial cell function *in vitro*, we first knocked down eNOS in cultured HUVECs and confirmed successful knockdown at transcript and protein levels in sieNOS-transfected endothelial cells (**Fig 1A and 1B**). We then evaluated cell proliferation in eNOS-knocked down and control endothelial cells using MTT assay. To our surprise, proliferation assay demonstrated significantly higher proliferation in sieNOS-transfected endothelial cells in comparison to control cells in a time-dependent manner at the studied 24, 48 and 72 hrs time-points (**Fig 1C**). We further validated our findings using automated Cytosmart cell-counter and another sieNOS# molecule, which also demonstrated significantly higher proliferation in the eNOS-knocked down endothelial cells (**S1A–S1D Fig**). Cell viability was also evaluated using Cytosmart cell-counter, but it was not affected by eNOS-knocked down at all three studied time points (**S1B Fig**). It is important to note that there are limited literatures investigating effect of knocking down eNOS on endothelial cells and there are almost no reports investigating the effect of eNOS-knock down on endothelial cell proliferation. However, the effect of eNOS-overexpression on cell proliferation has been investigated, which indirectly support our findings. Kader *et al*., reported that in bovine aortic endothelial cells eNOS overexpression inhibited endothelial cell proliferation [17]. Accordingly, we have previously also observed that *breast cancer gene 2*-deficiency in endothelial cells was associated with reduced eNOS expression and increased cell proliferation [14]. We then evaluated the effect of eNOS-knock down on p21 expression, which is a cyclin-dependent kinase inhibitor that promotes endothelial cell cycle arrest and inhibits proliferation [18]. Our transcript and protein data on p21 expression demonstrated reduced p21 expression in the eNOS-knocked down endothelial cells in comparison to control endothelial cells for all the studied (24, 48 and 72 hrs) time-points (**Fig 1E and 1F**). We also investigated the effect of eNOS-knockdown on the expression of endothelial Ki67, which is an endothelial cell proliferation marker [19]. Our Ki67-immunoblottng data demonstrated increased Ki67 expression following knockdown of eNOS in endothelial cells (**Fig 1F**). Knockdown of eNOS-associated reduced cell cycle inhibitor p21 expression and increased proliferation marker Ki67 expression, both support an increased endothelial cell proliferation following knockdown of eNOS in endothelial cells.

We also evaluated the other indices of endothelial function, such as cell migration and angiogenesis in the form of tube forming potential. Endothelial cell-specific knockdown of eNOS significantly inhibited tube forming potential in the form of reduced mesh area and

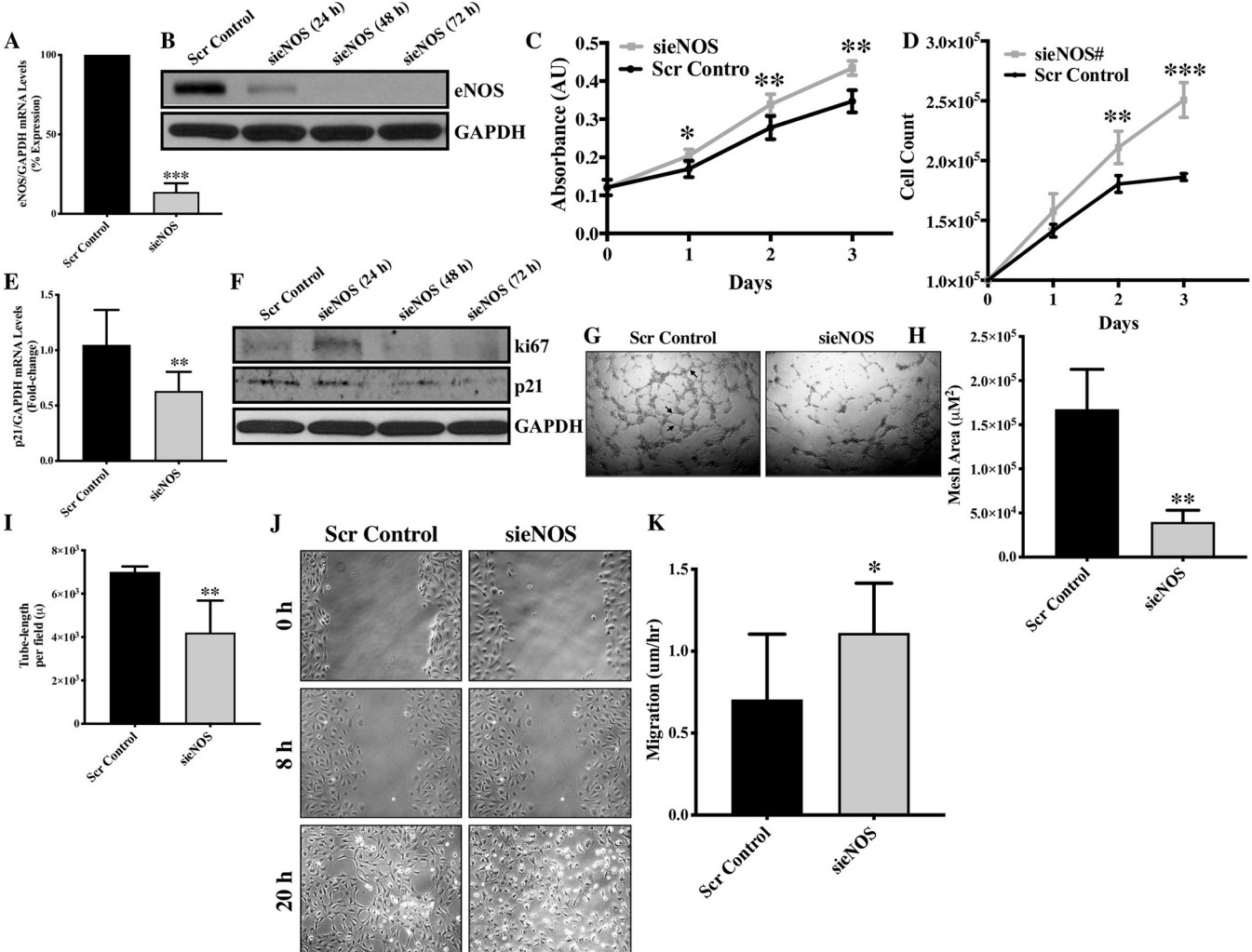

**Fig 1. Genetic knockdown of eNOS promotes proliferation and migration but inhibits formation of tube-like structures *in vitro* in HUVECs. (A)** HUVECs were transfected with sieNOS or scrambled control for 24, 48 and 72 hrs and the knockdown effect was confirmed by qPCR **(B)** western blot. **(C)** Endothelial cell proliferation was evaluated 24, 48 and 72 hrs post-transfection using proliferation kit and **(D)** by counting the cells using Cyto Smart cell counter. Each triplicate was counted twice, and average of all triplicates was calculated for each biological replicate. **(E)** Later, RNA was extracted 24 hrs post-transfection and qPCR for p21 and **(F)** protein was extracted 24, 48 and 72 hrs post-transfection and immunoblot was performed for ki67, p21 and GAPDH. **(G)** ENOS-knockdown and control HUVECs were seeded on Matrigel, and pictures were taken 6 hrs post-seeding (tubes are marked by arrows) and **(H)** then quantification for mesh area and **(I)** tube-length were performed. **(J)** Scratch assay was performed in sieNOS and scrambled control-transfected HUVECs and pictures were taken at 0, 8 and 20 hrs and **(K)** the cell migration was quantified. *p<0.05, **p<0.01, ***p<0.001 vs. Scr Control. N = 3–5 in triplicates.

tube-length in sieNOS-transfected endothelial cells **(Fig 1G–1I)**. This finding is in line with previous finding, where inhibiting eNOS expression in endothelial cells is shown to inhibit angiogenesis [20, 21] However, knockdown of eNOS significantly enhanced the migratory ability of endothelial cells **(Fig 1J and 1K)**.

HUVECs are very well characterized endothelial cells and have been used as representative endothelial cells to answer fundamental endothelial function-related questions [15, 16, 22], however, endothelial cells do display considerable functional and transcriptomic heterogeneity depending on their location [23]. To confirm that loss of eNOS-associated induction of proliferation and migration is not HUVEC-specific, we transfected Human Pulmonary Artery Endothelial Cells (HPAECSs) with sieNOS and confirmed successful eNOS silencing **(Fig 2A)**.

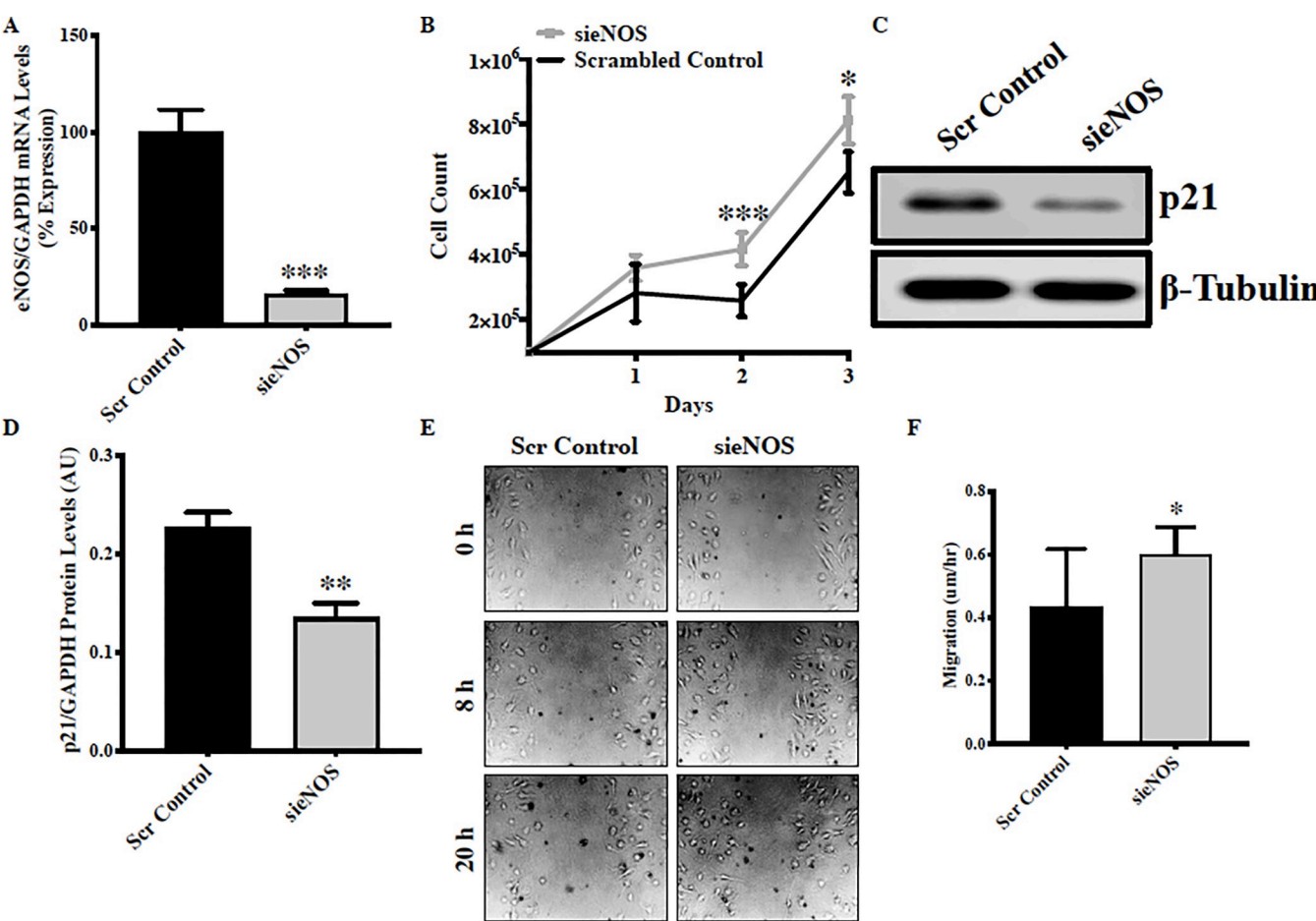

**Fig 2. Genetic knockdown of eNOS promotes proliferation and migration but inhibits formation of tube-like structures *in vitro* in HPAECs.** Human pulmonary artery endothelial cells (HPAECs) were transfected with sieNOS for 24, 48 and 72 hrs in HPAECs and **(A)** the knockdown effect was confirmed by qPCR. **(B)** Endothelial cell proliferation was evaluated 24, 48 and 72 hrs post-transfection using using Cyto Smart cell counter. Each triplicate was counted twice, and average of all triplicates was calculated for each biological replicate. **(C)** Protein was extracted 48 hrs post-transfection and immunoblot was performed for p21 and β-tubulin. **(D)** RNA was extracted 24 hrs post-transfection and qPCR for p21 was performed. **(E)** Scratch assay was performed in sieNOS and scrambled control-transfected HPAECs and pictures were taken at 0, 8 and 20 hrs **(F)** and the cell migration was quantified. $^*$p<0.05, $^{**}$p<0.01, $^{***}$p<0.001 *vs.* Scr Control. N = 3–5 in triplicates.

Similar to HUVECs, proliferation assay demonstrated significantly increased proliferation in eNOS-deficient in comparison to control HPAECs at all the three studied time-points (**Fig 2B**). Loss of eNOS-associated increased proliferation was again associated with reduced p21 expression in HPAECs (**Fig 2C and 2D**). Then, a scratch assay was performed on HPAECs and similar to HUVECs, an enhanced cell migration was observed in eNOS-deficient HPAECs (**Fig 2E and 2F**). This is contrasting to previous reports where eNOS pharmacologic inhibition by L-NAME is shown to inhibit endothelial cell migration[18, 19]. However, this contrasting finding may be attributed to the non-specificity of the inhibitor or the complex regulation of endothelial function by eNOS in a context-dependent manner. Interestingly, genetic transfer of eNOS in aortic smooth muscle cell is shown to inhibit migration [24], which indirectly support our data on knockdown of eNOS-associated induced cell migration. ENOS is an important regulator of VEGFa (vascular endothelial cell growth factor A), which further regulates angiogenesis and endothelial function by regulating VEGF/eNOS/AKT pathway [5]. We measured VEGFa transcript level in eNOS-knocked down endothelial cells, however VEGFa

transcript levels were unaffected by eNOS-knock down (**S1C Fig**). To further investigate the underlying mechanism of how knockdown of eNOS led to an increased cell proliferation, we evaluated the expression and activation level of mechanistic regulators of endothelial cell proliferation PI3-K/AKT and MAPK/ERK signaling pathway [25, 26]. AKT is a downstream target of phosphatidylinositol 3-kinase (PI3K) and the direct binding of PI3K and phosphorylated AKT leads to AKT activation. Subsequently, AKT can phosphorylate its downstream target including eNOS [27], which eventually leads to an increased NO release.

There is no literature investigating activation of AKT following eNOS knockdown. However, our immunoblotting data demonstrated a significant reduction in the activation of AKT in eNOS-knocked down endothelial cells in comparison to control cells (**Fig 3A–3C**). Initially, following 24 hrs of transfection, pAKT appeared to be up-regulated but this time does not coincide with the complete eNOS protein-loss following sieNOS transfection, so we quantified AKT activation 48-hrs post-transfection, which is associated with eNOS-protein loss (**Fig 1B**). Interestingly, eNOS-protein loss was associated with significantly reduced AKT activation, however, the mechanism behind eNOS loss-associated reduced AKT activation is unknown. It is also reported that inhibition of MAPK/ERK signaling pathway; such that deletion of ERK in primary endothelial cells results in decreased cell proliferation indicating an important role of ERK in endothelial cell proliferation [28]. Mechanistically, eNOS contains a motif that can be

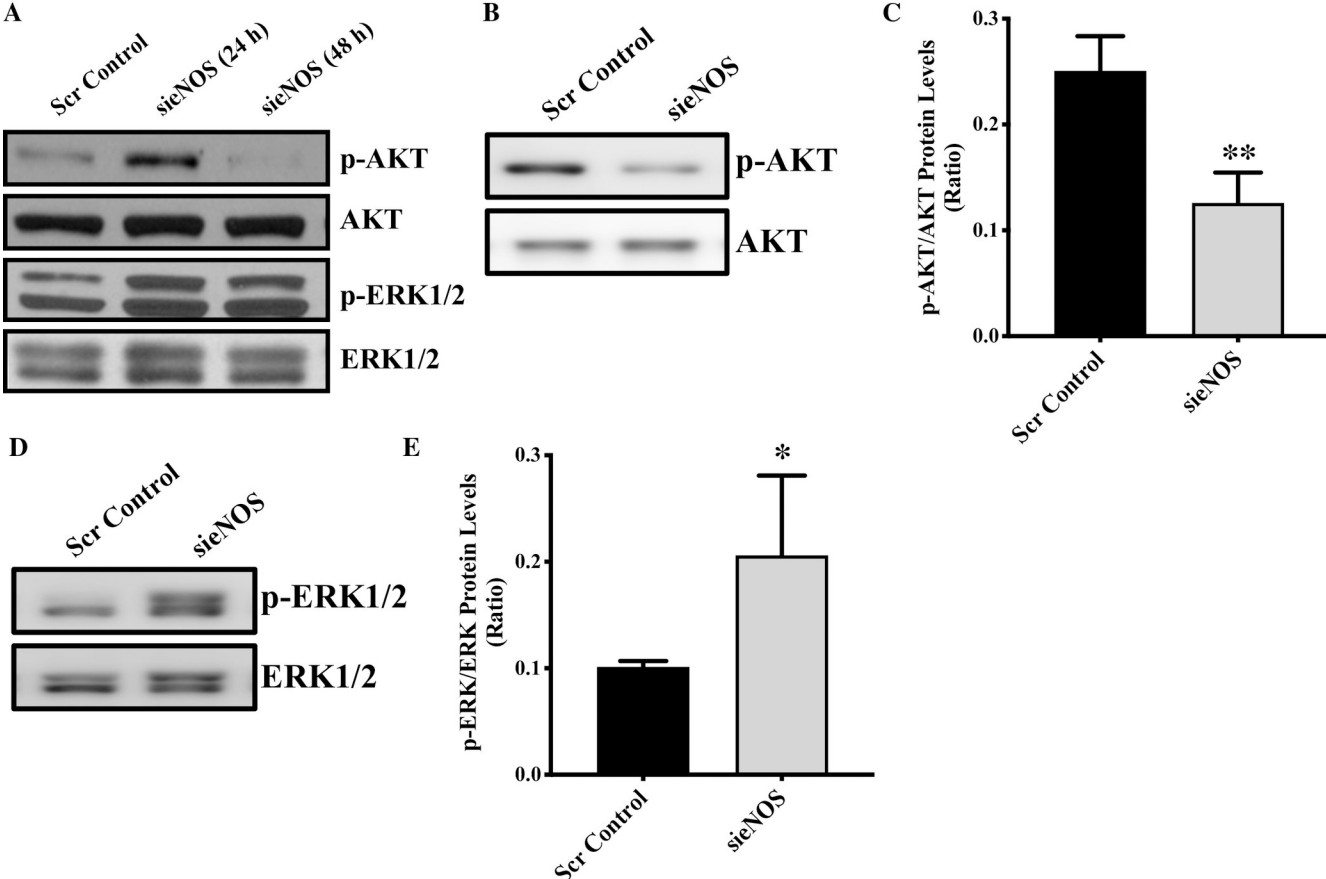

**Fig 3. Genetic knockdown of endothelial cell-specific eNOS inhibits AKT activation.** (**A**) HUVECs were transfected with sieNOS or scrambled control for 24 and 48 hrs. Total proteins were extracted 24 and 48 hrs post-transfection immunoblotting for p-AKT, AKT, p-ERK, ERK was performed. Immunoblots for (**B, C**) p-AKT, AKT, and (**D, E**) pERK1/2 and ERK1/2 were performed and quantified. *p<0.05, **p<0.01 *vs*. Scr Control. N = 3–5 in triplicates.

recognized by ERK, causing eNOS activation and thereby increased NO production [29]. Accordingly, we also measured ERK activation in eNOS-deficient endothelial cells. Our data show an increased trend towards increased ERK activation in eNOS-deficient endothelial cells (**Fig 3A, 3D and 3E**). It remains to be elucidated that how increased ERK-activation promotes proliferation in eNOS-deficient endothelial cells. Our findings for the first time show that knockdown of eNOS activates MAPK/ERK signaling pathway, which may potentially mediate enhanced endothelial cell proliferation in eNOS-deficient endothelial cells.

Given the unexpected nature of our findings about increased proliferation and migration but reduced angiogenesis in eNOS-deficient endothelial cells, we also evaluated proliferation, migration and angiogenesis following pharmacologic inhibition of eNOS using L-NAME. To our further surprise, we observed a dose-dependent inhibition of cell proliferation (**Fig 4A**) and tube forming potential (**Fig 4B–4D**) but increased cell migration (**Fig 4E and 4F**) in L-NAME-treated endothelial cells in comparison to vehicle-treated endothelial cells. Cell viability was not affected by L-NAME treatment (**S1D Fig**). However, this finding is in line with previous report, where L-NAME inhibited endothelial progenitor cells proliferation [20]. We

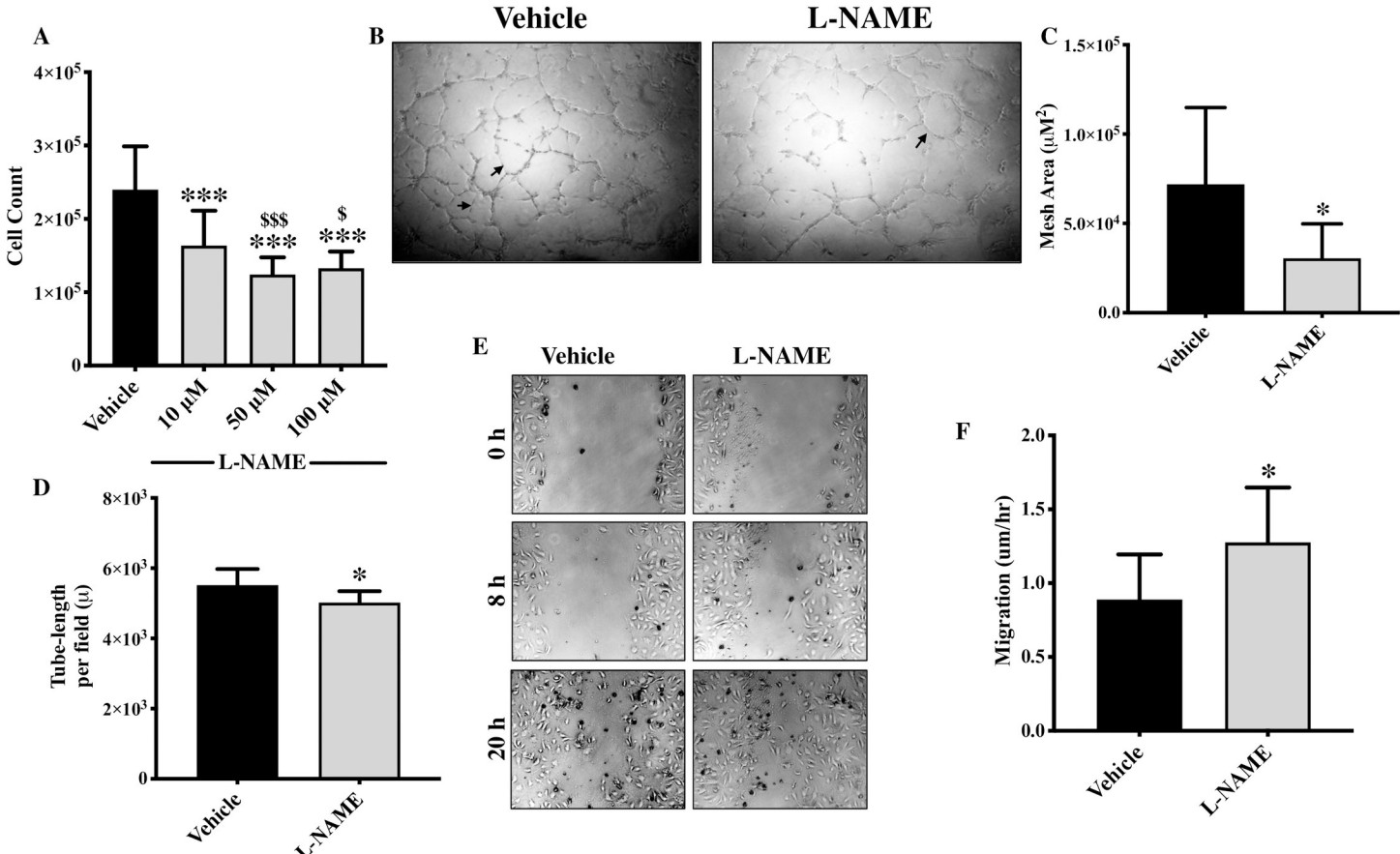

**Fig 4. Pharmacologic inhibition of eNOS-activity by L-NAME inhibits endothelial cell proliferation and formation of tube-like structures *in vitro*, but promotes endothelial cell migration.** HUVECs were seeded until they reached 60–70% confluency and were treated with different dose of L-NAME (10, 50 and 100uM). **(A)** Cells were collected and counted at each dose using Cyto Smart cell counter. Each triplicate was counted twice, and average of all triplicates was calculated for each biological replicate. **(B)** HUVECs were seeded on Matrigel, and pictures were taken 6 hrs post-seeding in the presence of L-NAME or vehicle (tubes are marked by arrows) and **(C)** then quantification for mesh area and **(D)** tube-length were performed. **(E)** HUVECs were seeded until they reached 70–80% confluency and were treated with L-NAME or vehicle. A scratch was made and pictures were taken at 0, 8 and 20 hrs and **(F)** the cell migration was quantified. $^{*}p<0.05$, $^{***}p<0.001$ vs. vehicle. #p<0.05, ###p<0.001 vs. 10μM L-NAME. N = 3–5 in triplicate.

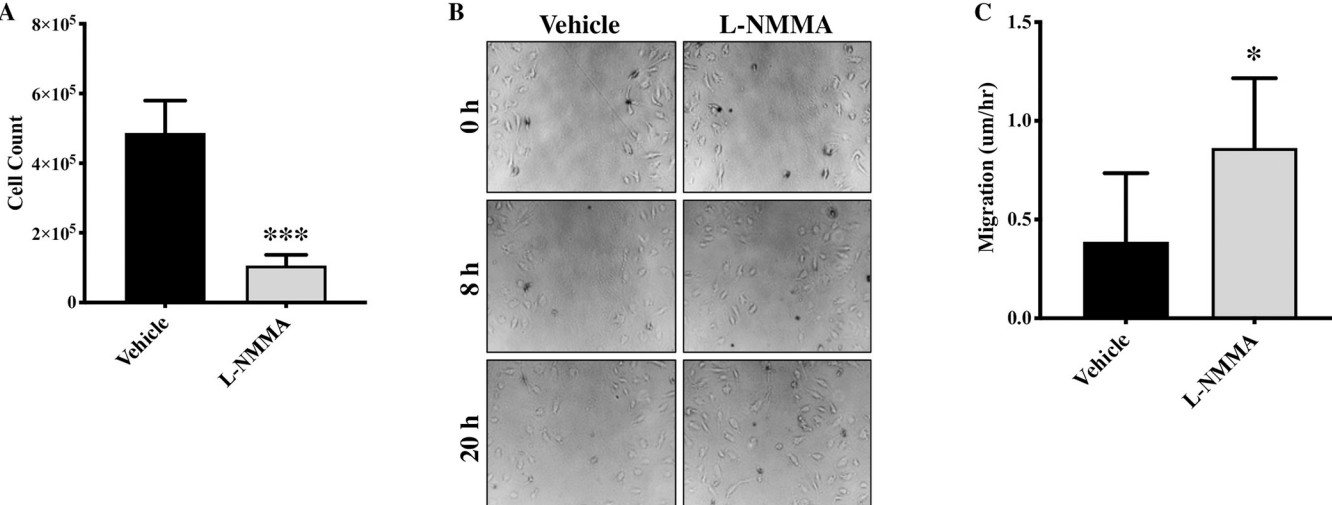

**Fig 5. Pharmacologic inhibition of eNOS-activity by L-NMMA inhibits endothelial cell proliferation but promotes endothelial cell migration. (A)** HUVECs were seeded until they reached 60% confluency and were treated with L-NMMA (50uM) or vehicle and 24 hrs later cells were collected and counted using Cyto Smart cell counter. Each triplicate was counted twice, and average of all triplicates was calculated for each biological replicate. **(B)** Cell migration assay was performed at 0, 8 and 20 hrs. HUVECs were seeded until they reached 70–80% confluency and were treated with L-NMMA or vehicle. A scratch was made and pictures were taken at 0, 8 and 20 hrs and **(C)** the cell migration was quantified. $^*p<0.05$, $^{***}p<0.001$ vs. vehicle. N = 3–5 in triplicate.

next used another and more-direct eNOS inhibitor; L-NMMA, which is the active compound in L-NAME [30]. L-NMMA-treatment to the endothelial cells replicated our finding from L-NAME, where L-NMMA treatment also inhibited cell proliferation (**Fig 5A**) and promoted cell migration (**Fig 5B and 5C**). The exact mechanism of eNOS-associated inhibition of cell proliferation needs to be further investigated, however, it appears that this phenotype is not directly arising from the inhibition of eNOS activity leading to NO production as it is not affected by L-NAME or L-NMMA but mechanisms associated with processes up-stream to eNOS activation. Genetic and pharmacologic inhibition of eNOS-associated migration and angiogenic potential indicate that these mechanisms are directly associated with eNOS activation and related down-stream processes such as NO production.

Inhibition of eNOS and NO; both are associated with development of cardiovascular diseases, and accordingly there are therapeutics that are in use and/or being developed to enhance eNOS expression/activity and NO production to treat cardiovascular diseases, such as myocardial infarction, cardiac hypertrophy, diastolic heart failure, arteriosclerosis, and hypertension in humans and in animal models [31, 32]. However, both, beneficial and detrimental effects of eNOS activation have also been reported [33]. Our findings about knockdown of eNOS on endothelial cell proliferation, migration, p21, PI3-K/AKT and MAPK/ERK signaling pathway adds further complexity to the eNOS expression/activation-mediated treatment strategy and warrants a more detailed investigation particularly in the settings where these factors are associated with the progression of the disease.

## Supporting information

**S1 Fig. (A)** HUVECs were transfected with either scrambled control or sieNOS# (5 nM each) for 24 hrs and RNA was extracted. The knockdown effect was confirmed by qPCR. $^{***}p<0.0001$ *vs*. Scr Control. **(B)** Cell viability was examined 24, 48 and 72 hrs post-transfection in HUVECs. Each triplicate was counted twice, and average of all triplicates was calculated for each biological replicate. **(C)** HUVECs were transfected with either scrambled control or

sieNOS for 48 hrs and RNA was extracted to perform qPCR for VEGFa. **(D)** Cell viability was evaluated following 24 hrs of vehicle or different dose of L-NAME treatment in HUVECs. Each triplicate was counted twice, and average of all triplicates was calculated for each biological replicate.
(TIF)

**S1 Raw images.**
(PPTX)

## Author Contributions

**Conceptualization:** Krishna K. Singh.

**Formal analysis:** Shuhan Bu, Hien C. Nguyen, Sepideh Nikfarjam, Shweta Singh, Krishna K. Singh.

**Funding acquisition:** Krishna K. Singh.

**Investigation:** Shuhan Bu, Krishna K. Singh.

**Methodology:** Krishna K. Singh.

**Project administration:** Krishna K. Singh.

**Resources:** Krishna K. Singh.

**Supervision:** Krishna K. Singh.

**Writing – original draft:** Shuhan Bu.

**Writing – review & editing:** Shuhan Bu, Hien C. Nguyen, Sepideh Nikfarjam, David C. R. Michels, Berk Rasheed, Sauraish Maheshkumar, Shweta Singh, Krishna K. Singh.

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
