## [Decision Letter · Decision Letter 0]

22 Feb 2022

PONE-D-22-01733Endothelial cell-specific Loss of eNOS Promotes ProliferationPLOS ONE

Dear Dr. Singh,

Thank you for submitting your manuscript to PLOS ONE. After careful consideration, we feel that it has merit but does not fully meet PLOS ONE’s publication criteria as it currently stands. Therefore, we invite you to submit a revised version of the manuscript that addresses the points raised during the review process.

The work is of interest since it proposes an original view of the potential role of eNOS on endothelial cells. However in the current form it seems yet preliminary and required many precisions and some experimental confirmations to be convincing. In particular it needs to take into account as underlined by one reviewer, in what context your hypothesis can take place and use the right tools for this aim (cell source, chemicals). Furthermore a point by point response to the concerns underlined by both reviewers is expected.

We look forward to receiving your revised manuscript.

Kind regards,

Alain-Pierre Gadeau, Ph.D

Academic Editor

PLOS ONE

Journal Requirements:

Reviewers' comments:

Reviewer's Responses to Questions

**Comments to the Author**

1. Is the manuscript technically sound, and do the data support the conclusions?

Reviewer #1: Partly

Reviewer #2: No

2. Has the statistical analysis been performed appropriately and rigorously? 

Reviewer #1: Yes

Reviewer #2: Yes

3. Have the authors made all data underlying the findings in their manuscript fully available?

Reviewer #1: No

Reviewer #2: Yes

4. Is the manuscript presented in an intelligible fashion and written in standard English?

Reviewer #1: Yes

Reviewer #2: No

5. Review Comments to the Author

Reviewer #1: Title refer only to proliferation, but Fig 1K also migration.

The main question of the paper is quite unclear : Do the authors through proliferation and migration sought to modelize macrovascular endothelial cell healing, or do they sought to modelize angiogenesis ?

In addition, they refer to eNOS expression in SMC (ref 19) where the mechanisms are totaly different and which pathophysiological meaning is unclear.

FInally, they report rather preliminary results in nature, and acknowledge that at the end of the introduction « Our findings confirm that eNOS regulate endothelial function by directly controlling endothelial cell proliferation, and nonspecificity of L-NAME is responsible for inhibition of endothelial cell proliferation; both warranting further investigation.. ». Indeed, these results are very descriptive and the amount of limited to 2 Figures.

Thus, the apparent « discrepancy » between gene expression inhibition and eNOS actinity inhibition should be investigated more in deepth.

Major points :

1) INTRO :

a) « PKG increases extracellular concentration of calcium by facilitating the reuptake of cytosolic calcium into the sarcoplasmic reticulum and opening of calcium-activated potassium channels [5].

Ref 5 : Nitric oxide decreases [Ca2+]i in vascular smooth muscle by inhibition of the calcium current. Cell Calcium 1994. Is not appropriate as i) it demonstrate in SMC the opposite and ii) the roles of Calcium in SMC and in endothelial cells are very different and cannot be extrapolated.

2) METHODS :

a) Proliferation and Scratch Assay: the experimental conditions of cell culture should be better defined.

b) L-NAME is in vivo desestherified to L-NA, that is in fact the active coumpound. Thus,

Experiments should be done with L-NA and L-NMMA in order to minimize the pharmacological biais and to see if the results if these 3 complementary apporaches do concur. This would allow to attenuate the limitation indicated in the introduction : « non-specificity of eNOS pharmacologic inhibitor L-NAME « .

c) How many different sieNOS were used ? At least 3 should minimize the possible off-target effects.

Minor points :

Abstract : « On the contrary,.. » should be better explained : « at the level of the enzyme activity ».

Reviewer #2: The authors in their manuscript "Endothelial cell-specific Loss of eNOS Promotes Proliferation" attempted to describe function of eNOS as a negative regulator of cell proliferation. The study has a few major issues:

1. Authors should use correct term, authors claim that they are 'silencing' eNOS gene. Here, the word 'silencing' is used incorrectly, it should be replaced with 'siRNA mediated knockdown' or simply 'knockdown'. Genetic silencing, by definition, means traditional gene knockout using homologous recombination technology using the embryonic stem cells.

2. Can authors provide evidence that eNOS is a negative regulator of cell proliferation? For example, by over-expressing eNOS-cDNA in endothelial cells, which should increase the levels of cell cycle inhibitor such as p21, p27, p53 and down-regulation of Cyclin-D1. These can be done using cultured endothelial cells, at least two different primary endothelial cells must be used. The HUVECs at passage 4-6 are likely not quiescent, therefore, it would be more appropriate to use microvascular endothelial cells such as from lungs and heart. My fear is that the HUVECs may be displaying inappropriate adaptive response, in response to eNOS-knockdown.

3. Can authors normalize the activity of AKT (pAKT) in endothelial cells that received eNOS siRNA? Thereby rescue/restore the normal EC phenotype? Alternatively, can authors add back eNOS-cDNA to restore normal phenotype?

4. Figure 1L: Quantification of Western Blot is required.

5. The migration assay (wound healing) done over a period of 20 hours could include cell proliferation as well. Did author add BrdU during cell migration assay? This data is unclear.

6. In the Figure legend, can authors mention as to how many times experiments were repeated? Please explain N value clearly.

7. 'Tube formation' assay is misleading. Can authors indicate where the tubes are?

8. Methods are described inadequately.

Minor concerns:

1. 'matrigel' should be spelled Matrigel, where 'M' should be capitalized.

2. The manuscript could benefit from check on English grammar and clarity.

6. PLOS authors have the option to publish the peer review history of their article (what does this mean?). If published, this will include your full peer review and any attached files.

Reviewer #1: **Yes: **JF ARNAL

Reviewer #2: No

---

## [Author Response · Author response to Decision Letter 0]

22 Jul 2022

Reviewer 1

Comment #1 - Title refer only to proliferation, but Fig 1K also migration.

Response: Many thanks for your thoughtful comments. In cultured endothelial cells, loss of eNOS promoted cell proliferation and migration but inhibited tube-forming potential. Accordingly, to have more conclusive title, the revised title of the manuscript is that “Endothelial cell-specific Loss of eNOS Differentially Affects Endothelial Function”.

Comment #2: The main question of the paper is quite unclear: Do the authors through proliferation and migration sought to modelize macrovascular endothelial cell healing, or do they sought to modelize angiogenesis? In addition, they refer to eNOS expression in SMC (ref 19) where the mechanisms are totally different, and which pathophysiological meaning is unclear.?

Response: We mainly aimed to investigate the effect of loss of eNOS on in vitro measures of endothelial function such as proliferation, migration and tube formation, which contribute towards both; endothelial cell healing and angiogenesis. The statement regarding eNOS expression in SMC has been deleted from the revised manuscript.

Comment #3: Finally, they report rather preliminary results in nature, and acknowledge that at the end of the introduction “Our findings confirm that eNOS regulate endothelial function by directly controlling endothelial cell proliferation, and non-specificity of L-NAME is responsible for inhibition of endothelial cell proliferation; both warranting further investigation”. Indeed, these results are very descriptive and the amount of limited to 2 Figures. Thus, the apparent « discrepancy » between gene expression inhibition and eNOS activity inhibition should be investigated more in depth.

Response: We agree with the reviewer that to some extent the results are descriptive in nature; however, it confirms the effect of loss of eNOS on endothelial function in vitro, which has not been clearly reported so far, and that is the focus of the present study. We have plans to investigate the mechanisms behind observed effect, which will be part of our next manuscript. Explanation regarding non-specificity of L-NAME has been deleted from the revised manuscript. We believe that the differential effect of eNOS on endothelial function is observed due to mechanisms up-stream (before eNOS activation) and down-stream (after eNOS activation) to eNOS. 

Comment #5: PKG increases extracellular concentration of calcium by facilitating the reuptake of cytosolic calcium into the sarcoplasmic reticulum and opening of calcium-activated potassium channels [5]. Ref 5 : Nitric oxide decreases [Ca2+] in vascular smooth muscle by inhibition of the calcium current. Cell Calcium 1994. Is not appropriate as i) it demonstrate in SMC the opposite and ii) the roles of Calcium in SMC and in endothelial cells are very different and cannot be extrapolated.

Response: We agree with your thoughtful comment and have deleted the original reference and added new references in Page 3 paragraph 1 of the revised manuscript. 

Comment #6a: Proliferation and Scratch Assay: the experimental conditions of cell culture should be better defined.

Response: Thank you for your comment, the experimental conditions have now been further clarified in page 6, paragraph 1 and 2 of the revised manuscript. The migration assay was conducted as described in “An introduction to the wound healing assay using live-cell microscopy” (PMID: 25482647). 

Comment #6b: L-NAME is in vivo desestherified to L-NA, that is in fact the active coumpound. Thus, Experiments should be done with L-NA and L-NMMA in order to minimize the pharmacological bias and to see if the results if these 3 complementary apporaches do concur. This would allow to attenuate the limitation indicated in the introduction : « non-specificity of eNOS pharmacologic inhibitor L-NAME.

Response: Many thanks for your suggestion. Now, we have included data on proliferation and migration following L-NMMA, which is similar to L-NAME data on HUVECS proliferation, migration and tube-formation (Please see Fig. 5 in the revised manuscript).

Comment #6c: How many different sieNOS were used? At least 3 should minimize the possible off-target effects.

Response: We used two sieNOS (sieNOS and sieNOS#) molecule and then measured proliferation following silencing. However, we observed the similar reduction in proliferation following transfection with both siRNA molecules (Please see supplementary figure 1 and figure 1D). 

Comment #7: Abstract : “On the contrary,.. “ should be better explained : “at the level of the enzyme activity”.

Response: Abstract is revised as suggested.

Reviewer 2

Comment #1: Authors should use correct term, authors claim that they are 'silencing' eNOS gene. Here, the word 'silencing' is used incorrectly, it should be replaced with 'siRNA mediated knockdown' or simply 'knockdown'. Genetic silencing, by definition, means traditional gene knockout using homologous recombination technology using the embryonic stem cells.

Response: Thank you for your comment, it has now been corrected in the manuscript.

Comment #2: Can authors provide evidence that eNOS is a negative regulator of cell proliferation? For example, by over-expressing eNOS-cDNA in endothelial cells, which should increase the levels of cell cycle inhibitor such as p21, p27, p53 and down-regulation of Cyclin-D1. These can be done using cultured endothelial cells, at least two different primary endothelial cells must be used. The HUVECs at passage 4-6 are likely not quiescent, therefore, it would be more appropriate to use microvascular endothelial cells such as from lungs and heart. My fear is that the HUVECs may be displaying inappropriate adaptive response, in response to eNOS-knockdown. 

Response: Thank you for the thoughtful comment. We have followed your suggestion and silenced eNOS in Human Pulmonary Artery Endothelial Cell (HPAECs) and confirmed downregulation of eNOS. Knockdown of eNOS in HPAECS was associated with increased proliferation, downregulation of p21 and enhanced migration as observed in eNOS-deficient HUVECs (Please see Fig. 2 in the revised manuscript). We agree that eNOS-overexpression will very well complement our findings. Accordingly, we have future plans but that will form the basis of another manuscript.

Comment #3: Can authors normalize the activity of AKT (pAKT) in endothelial cells that received eNOS siRNA? Thereby rescue/restore the normal EC phenotype? Alternatively, can authors add back eNOS-cDNA to restore normal phenotype?

Response: As suggested, we have normalized the AKT activity. Please see figure 3B and C.

Comment #4. Figure 1L: Quantification of Western Blot is required. 

Response: As suggested, we have quantified AKT and ERK protein levels. Please see figure 3B-E.

Comment #5: The migration assay (wound healing) done over a period of 20 hours could include cell proliferation as well. Did author add BrdU during cell migration assay? This data is unclear.

Response: Thanks for your comment. We incubated cells in DMEM supplemented with 1%FBS to ensure there is only migration but not proliferation. Method is now clarified in page 6 paragraph 2. 

Comment #6: In the Figure legend, can authors mention as to how many times experiments were repeated? Please explain N value clearly.

Response: Thank you for the comment. We have repeated the experiments at least 3 times in triplicates and the N values are now added in the figure legend.

Comment #7: 'Tube formation' assay is misleading. Can authors indicate where the tubes are?

Response: Thanks for pointing it out. As suggested, tubes are marked in the figure 1G of the revised manuscript.

Comment #8: Methods are described inadequately.

Response: Thank you for the comment, we have added more details to the methods section. 

Comment #9: 'matrigel' should be spelled Matrigel, where 'M' should be capitalized.

Response: Thanks for the reminder, it has now been corrected.

Comment #10: The manuscript could benefit from check on English grammar and clarity.

Response: Thanks for the suggestion. We have now checked the manuscript for grammar and clarity.

---

## [Decision Letter · Decision Letter 1]

15 Aug 2022

PONE-D-22-01733R1Endothelial Cell-Specific Loss of eNOS Differentially Affects Endothelial FunctionPLOS ONE

Dear Dr. Singh,

Thank you for submitting your manuscript to PLOS ONE. After careful consideration, we feel that it has merit but does not fully meet PLOS ONE’s publication criteria as it currently stands. Therefore, we invite you to submit a revised version of the manuscript that addresses the points raised during the review process. Please submit your revised manuscript by Sep 29 2022 11:59PM. If you will need more time than this to complete your revisions, please reply to this message or contact the journal office at plosone@plos.org. Please include the following items when submitting your revised manuscript:A rebuttal letter that responds to each point raised by the academic editor and reviewer(s). You should upload this letter as a separate file labeled 'Response to Reviewers'.A marked-up copy of your manuscript that highlights changes made to the original version. You should upload this as a separate file labeled 'Revised Manuscript with Track Changes'.An unmarked version of your revised paper without tracked changes. You should upload this as a separate file labeled 'Manuscript'.If applicable, we recommend that you deposit your laboratory protocols in protocols.io to enhance the reproducibility of your results. Protocols.io assigns your protocol its own identifier (DOI) so that it can be cited independently in the future. For instructions see: https://journals.plos.org/plosone/s/submission-guidelines#loc-laboratory-protocols. Additionally, PLOS ONE offers an option for publishing peer-reviewed Lab Protocol articles, which describe protocols hosted on protocols.io. Read more information on sharing protocols at https://plos.org/protocols?utm_medium=editorial-email&utm_source=authorletters&utm_campaign=protocols.

We look forward to receiving your revised manuscript.

Kind regards,

Jeffrey S Isenberg, MD, MPH

Academic Editor

PLOS ONE

Journal Requirements:

Additional Editor Comments (if provided):

The Reviewers felt that the revised manuscript was much improved, and the authors are thanked for this. Some concern still was found regarding terminology. See below and please address this. As well, a Reviewer requested an effort be made to improve the overall quality of the figures.

[However, the use of terminology remains confusing, e.g., tube formation. If authors claim that these are indeed tubes (Figure 4B, indicated by arrows), the Matrigel must be fixed with paraformaldehyde (4% PFA) and embedded in paraffin, make cross thin section, thereafter, stain with H&E and show lumen (vacuole/empty space). In other words, "lumen formation" is alternatively called "tube formation".]

Reviewers' comments:

Reviewer's Responses to Questions

**Comments to the Author**

1. If the authors have adequately addressed your comments raised in a previous round of review and you feel that this manuscript is now acceptable for publication, you may indicate that here to bypass the “Comments to the Author” section, enter your conflict of interest statement in the “Confidential to Editor” section, and submit your "Accept" recommendation.

Reviewer #1: All comments have been addressed

Reviewer #2: All comments have been addressed

2. Is the manuscript technically sound, and do the data support the conclusions?

Reviewer #1: Yes

Reviewer #2: Yes

3. Has the statistical analysis been performed appropriately and rigorously? 

Reviewer #1: Yes

Reviewer #2: Yes

4. Have the authors made all data underlying the findings in their manuscript fully available?

Reviewer #1: Yes

Reviewer #2: Yes

5. Is the manuscript presented in an intelligible fashion and written in standard English?

Reviewer #1: Yes

Reviewer #2: Yes

6. Review Comments to the Author

Reviewer #1: Seems fine no additional comment Seems fine no additional comment Seems fine no additional comment Seems fine no additional comment

Reviewer #2: The authors have addressed most of my questions and concerns.

However, the use of terminology remains confusing, e.g., tube formation. If authors claim that these are indeed tubes (Figure 4B, indicated by arrows), the Matrigel must be fixed with paraformaldehyde (4% PFA) and embedded in paraffin, make cross thin section, thereafter stain with H&E and show lumen (vacuole/empty space). In other words, "lumen formation" is alternatively called "tube formation".

7. PLOS authors have the option to publish the peer review history of their article (what does this mean?). If published, this will include your full peer review and any attached files.

Reviewer #1: **Yes: **Jean Francois ARNAL

Reviewer #2: No

---

## [Author Response · Author response to Decision Letter 1]

26 Aug 2022

Reviewer #1: Seems fine no additional comment Seems fine no additional comment Seems fine no additional comment Seems fine no additional comment

Response: Thank you very much. 

Reviewer#2: The authors have addressed most of my questions and concerns.

However, the use of terminology remains confusing, e.g., tube formation. If authors claim that these are indeed tubes (Figure 4B, indicated by arrows), the Matrigel must be fixed with paraformaldehyde (4% PFA) and embedded in paraffin, make cross thin section, thereafter stain with H&E and show lumen (vacuole/empty space). In other words, "lumen formation" is alternatively called "tube formation".

Response: Thank you for your insightful comment. We agree that endothelial cells link to each other through formation of junctional complexes upon stimulation of angiogenic signals which lead to formation of lumen, and fixation will help visualize the lumen. We have now changed the “tube formation” to “formation of tube-like structures”.

---

## [Editor Report · Decision Letter 2]

30 Aug 2022

Endothelial Cell-Specific Loss of eNOS Differentially Affects Endothelial Function

PONE-D-22-01733R2

Dear Dr. Singh,

We’re pleased to inform you that your manuscript has been judged scientifically suitable for publication and will be formally accepted for publication once it meets all outstanding technical requirements.

Kind regards,

Jeffrey S Isenberg, MD, MPH

Academic Editor

PLOS ONE

Additional Editor Comments (optional):

The authors are thanked for the minor adjustment in terminology. The manuscript is acceptable for publication.
---

## [Editor Report · Acceptance letter]

15 Sep 2022

PONE-D-22-01733R2 

Endothelial Cell-specific Loss of eNOS Differentially Affects Endothelial Function 

Dear Dr. Singh:

I'm pleased to inform you that your manuscript has been deemed suitable for publication in PLOS ONE. Congratulations! Your manuscript is now with our production department. 

Kind regards, 

on behalf of

Dr. Jeffrey S Isenberg 

Academic Editor

PLOS ONE